# Randomized Ensembled Double Q-Learning: Learning Fast Without a Model

**Xinyue Chen**[1*]  **Che Wang**[1,2*]  **Zijian Zhou**[1*]  **Keith Ross**[1,2†]

[1] New York University Shanghai
[2] New York University

## Abstract

Using a high Update-To-Data (UTD) ratio, model-based methods have recently achieved much higher sample efficiency than previous model-free methods for continuous-action DRL benchmarks. In this paper, we introduce a simple model-free algorithm, Randomized Ensembled Double Q-Learning (REDQ), and show that its performance is just as good as, if not better than, a state-of-the-art model-based algorithm for the MuJoCo benchmark. Moreover, REDQ can achieve this performance using fewer parameters than the model-based method, and with less wall-clock run time. REDQ has three carefully integrated ingredients which allow it to achieve its high performance: (i) a UTD ratio $\gg 1$; (ii) an ensemble of Q functions; (iii) in-target minimization across a random subset of Q functions from the ensemble. Through carefully designed experiments, we provide a detailed analysis of REDQ and related model-free algorithms. To our knowledge, REDQ is the first successful model-free DRL algorithm for continuous-action spaces using a UTD ratio $\gg 1$.

## 1 Introduction

Recently, model-based methods in continuous action space domains have achieved much higher sample efficiency than previous model-free methods. Model-based methods often attain higher sample efficiency by using a high Update-To-Data (UTD) ratio, which is the number of updates taken by the agent compared to the number of actual interactions with the environment. For example, Model-Based Policy Optimization (MBPO) (Janner et al., 2019), is a state-of-the-art model-based algorithm which updates the agent with a mix of real data from the environment and "fake" data from its model, and uses a large UTD ratio of 20-40. Compared to Soft-Actor-Critic (SAC), which is model-free and uses a UTD of 1, MBPO achieves much higher sample efficiency in the OpenAI MuJoCo benchmark (Todorov et al., 2012; Brockman et al., 2016). This raises the question of whether it is also possible to achieve such high performance without a model?

In this paper, we introduce a simple model-free algorithm called Randomized Ensemble Double Q learning (REDQ), and show that its performance is just as good as, if not better than, MBPO. The result indicates, that at least for the MuJoCo benchmark, simple model-free algorithms can attain the performance of current state-of-the-art model-based algorithms. Moreover, REDQ can achieve this performance using fewer parameters than MBPO, and with less wall-clock run time.

Like MBPO, REDQ employs a UTD ratio that is $\gg 1$, but unlike MBPO it is model-free, has no roll outs, and performs all updates with real data. In addition to using a UTD ratio that is $\gg 1$, it has two other carefully integrated ingredients: an ensemble of Q functions; and in-target minimization across a *random subset* of Q functions from the ensemble.

Through carefully designed experiments, we provide a detailed analysis of REDQ. We introduce the metrics of average Q-function bias and standard deviation (std) of Q-function bias. Our results show that using ensembles with in-target minimization reduces the std of the Q-function bias to close to

---

[*]Equal contribution, in alphabetical order.
[†]Correspondence to: Keith Ross <keithwross@nyu.edu>.

zero for most of training, even when the UTD is very high. Furthermore, by adjusting the number of randomly selected Q-functions for in-target minimization, REDQ can control the average Q-function bias. In comparison with standard ensemble averaging and with SAC with a higher UTD, REDQ has much lower std of Q-function bias while maintaining an average bias that is negative but close to zero throughout most of training, resulting in significantly better learning performance. We perform an ablation study, and show that REDQ is very robust to choices of hyperparameters, and can work well with a small ensemble and a small number of Q functions in the in-target minimization. We also provide a theoretical analysis, providing additional insights into REDQ. Finally, we consider combining the REDQ algorithm with an online feature extractor network (OFENet) (Ota et al., 2020) to further improve performance, particularly for the more challenging environments Ant and Humanoid. We achieve more than 7x the sample efficiency of SAC to reach a score of 5000 for both Ant and Humanoid. In Humanoid, REDQ-OFE also greatly outperforms MBPO, reaching a score of 5000 at 150K interactions, which is 3x MBPO's score at that point.

To ensure our comparisons are fair, and to ensure our results are reproducible (Henderson et al., 2018; Islam et al., 2017; Duan et al., 2016), we provide open source code[1]. For all algorithmic comparisons, we use the same codebase (except for MBPO, for which we use the authors' code).

## 2   RANDOMIZED ENSEMBLED DOUBLE Q-LEARNING (REDQ)

Janner et al. (2019) proposed Model-Based Policy Optimization (MBPO), which was shown to be much more sample efficient than popular model-free algorithms such as SAC and PPO for the MuJoCo environments. MBPO learns a model, and generates "fake data" from its model as well as "real data" through environment interactions. It then performs parameter updates using both the fake and the real data. One of the distinguishing features of MBPO is that it has a UTD ratio $\gg 1$ for updating its Q functions, enabling MBPO to achieve high sample efficiency.

We propose Randomized Ensembled Double Q-learning (REDQ), a novel model-free algorithm whose sample-efficiency performance is just as good as, if not better than, the state-of-the-art model-based algorithm for the MuJoCo benchmark. The pseudocode for REDQ is shown in Algorithm 1. REDQ can be used with any standard off-policy model-free algorithm, such as SAC (Haarnoja et al., 2018b), SOP (Wang et al., 2019), TD3 (Fujimoto et al., 2018), or DDPG (Lillicrap et al., 2015). For the sake of concreteness, we use SAC in Algorithm 1. REDQ has the following key components: $(i)$ To improve sample efficiency, the UTD ratio $G$ is much greater than one; $(ii)$ To reduce the variance in the Q-function estimate, REDQ uses an ensemble of $N$ Q-functions, with each Q-function randomly and independently initialized but updated with the same target; $(iii)$ To reduce over-estimation bias, the target for the Q-function includes a minimization over a *random subset* $\mathcal{M}$ of the $N$ Q-functions. The size of the subset $\mathcal{M}$ is kept fixed, and is denoted as $M$, and is referred to as the *in-target minimization parameter*. Since our default choice for $M$ is $M = 2$, we refer to the algorithm as Randomized Ensembled *Double* Q-learning (REDQ).

REDQ shares some similarities with Maxmin Q-learning (Lan et al., 2020), which also uses ensembles and also minimizes over multiple Q-functions in the target. However, Maxmin Q-learning and REDQ have many differences, e.g., Maxmin Q-learning minimizes over the full ensemble in the target, whereas REDQ minimizes over a random subset of Q-functions. Unlike Maxmin Q-learning, REDQ controls over-estimation bias and variance of the Q estimate by separately setting $M$ and $N$. REDQ has many possible variations, some of which are discussed in the ablation section.

REDQ has three key hyperparameters, $G$, $N$, and $M$. When $N = M = 2$ and $G = 1$, then REDQ simply becomes the underlying off-policy algorithm such as SAC. When $N = M > 2$ and $G = 1$, then REDQ is similar to, but not equivalent to, Maxmin Q-learning (Lan et al., 2020). In practice, we find $M = 2$ works well for REDQ, and that a wide range of values around $N = 10$ and $G = 20$ work well. To our knowledge, REDQ is the first successful model-free DRL algorithm for continuous-action spaces using a UTD ratio $G \gg 1$.

---

[1]Code and implementation tutorial can be found at: `https://github.com/watchernyu/REDQ`

---

**Algorithm 1** Randomized Ensembled Double Q-learning (REDQ)

---

1: Initialize policy parameters $\theta$, $N$ Q-function parameters $\phi_i$, $i = 1, \ldots, N$, empty replay buffer $\mathcal{D}$. Set target parameters $\phi_{\text{targ},i} \leftarrow \phi_i$, for $i = 1, 2, \ldots, N$
2: **repeat**
3:     Take one action $a_t \sim \pi_\theta(\cdot|s_t)$. Observe reward $r_t$, new state $s_{t+1}$.
4:     Add data to buffer: $\mathcal{D} \leftarrow \mathcal{D} \cup \{(s_t, a_t, r_t, s_{t+1})\}$
5:     **for** $G$ updates **do**
6:         Sample a mini-batch $B = \{(s, a, r, s')\}$ from $\mathcal{D}$
7:         Sample a set $\mathcal{M}$ of $M$ distinct indices from $\{1, 2, \ldots, N\}$
8:         Compute the Q target $y$ (same for all of the $N$ Q-functions):

$$y = r + \gamma \left( \min_{i \in \mathcal{M}} Q_{\phi_{\text{targ},i}} (s', \tilde{a}') - \alpha \log \pi_\theta (\tilde{a}' \mid s') \right), \quad \tilde{a}' \sim \pi_\theta (\cdot \mid s')$$

9:         **for** $i = 1, \ldots, N$ **do**
10:           Update $\phi_i$ with gradient descent using

$$\nabla_\phi \frac{1}{|B|} \sum_{(s,a,r,s') \in B} (Q_{\phi_i}(s,a) - y)^2$$

11:           Update target networks with $\phi_{\text{targ},i} \leftarrow \rho \phi_{\text{targ},i} + (1-\rho)\phi_i$
12:     Update policy parameters $\theta$ with gradient ascent using

$$\nabla_\theta \frac{1}{|B|} \sum_{s \in B} \left( \frac{1}{N} \sum_{i=1}^{N} Q_{\phi_i} (s, \tilde{a}_\theta(s)) - \alpha \log \pi_\theta (\tilde{a}_\theta(s)|s) \right), \quad \tilde{a}_\theta(s) \sim \pi_\theta(\cdot \mid s)$$

---

## 2.1 EXPERIMENTAL RESULTS FOR REDQ

We now provide experimental results for REDQ and MBPO for the four most challenging MuJoCo environments, namely, Hopper, Walker2d, Ant, and Humanoid. We have taken great care to make a fair comparison of REDQ and MBPO. The MBPO results are reproduced using the author's open source code, and we use the hyperparameters suggested in the MBPO paper, including $G = 20$. We obtain MBPO results similar to those reported in the MBPO paper. For REDQ, we use $G = 20$, $N = 10$, and $M = 2$ for all environments. We use the evaluation protocol proposed in the MBPO paper. Specifically, after every epoch we run one test episode with the current policy and record the performance as the undiscounted sum of all the rewards in the episode. A more detailed discussion on hyperparameters and implementation details is given in the Appendix.

Figure 1 shows the training curves for REDQ, MBPO, and SAC. For each algorithm, we plot the average return of 5 independent trials as the solid curve, and plot the standard deviation across 5 seeds as the transparent shaded region. For each environment, we train each algorithm for exactly the same number of environment interactions as done in the MBPO paper. Figure 1 shows that both REDQ and MBPO learn much faster than SAC, with REDQ performing somewhat better than MBPO on the whole. In particular, REDQ learns significantly faster for Hopper, and has somewhat better asymptotic performance for Hopper, Walker2d, and Humanoid. A more detailed performance comparison is given in the Appendix, where it is shown that, averaging across the environments, REDQ performs 1.4x better than MBPO half-way through training and 1.1x better at the end of training. These results taken together are perhaps counter-intuitive. They show that a simple model-free algorithm can achieve as good or better sample-efficiency performance as the state-of-the-art model-based algorithm for the MuJoCo environments.

Does REDQ achieve its sample efficiency using more computational resources than MBPO? We now compare the number of parameters used in REDQ and MBPO. With REDQ, for each Q network and the policy network, we use a multi-layer perceptron with two hidden layers, each with 256 units. For MBPO, we use the default network architectures for the Q networks, policy network, and model ensembles (Janner et al., 2019). The Appendix provides a table comparing the number of parameters: REDQ uses fewer parameters than MBPO for all four environments, specifically, between

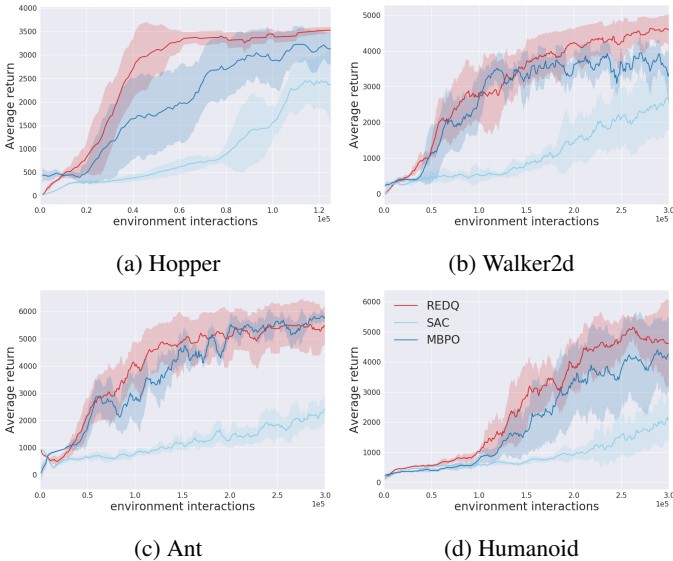

(a) Hopper

(b) Walker2d

(c) Ant

(d) Humanoid

Figure 1: REDQ compared to MBPO and SAC. Both REDQ and MBPO use $G = 20$.

26% and 70% as many parameters depending on the environment. Additionally, we measured the runtime on a 2080-Ti GPU and found that MBPO roughly takes 75% longer. In summary, the results in this section show that the model-free algorithm REDQ is not only at least as sample efficient as MBPO, but also has fewer parameters and is significantly faster in terms of wall-clock time.

## 3 WHY DOES REDQ SUCCEED WHEREAS OTHERS FAIL?

REDQ is a simple model-free algorithm that matches the performance of a state-of-the-art model-based algorithm. Key to REDQ's sample efficiency is using a UTD $\gg 1$. Why is it that SAC and ordinary ensemble averaging (AVG) cannot do as well as REDQ by simply increasing the UTD?

To address these questions, let $Q^\pi(s, a)$ be the action-value function for policy $\pi$ using the standard infinite-horizon discounted return definition. Let $Q_\phi(s, a)$ be an estimate of $Q^\pi(s, a)$, which is defined as the average of $Q_{\phi_i}(s, a)$, $i = 1, \ldots, N$, when using an ensemble. We define the bias of an estimate at state-action pair $(s, a)$ to be $Q_\phi(s, a) - Q^\pi(s, a)$. We are primarily interested in the accuracy of $Q_\phi(s, a)$ over the state-action distribution of the current policy $\pi$. To quantitatively analyze how estimation error accumulates in the training process, we perform an analysis that is similar to previous work (Van Hasselt et al., 2016; Fujimoto et al., 2018), but not exactly the same. We run a number of analysis episodes from different random initial states using the current policy $\pi$. For each state-action pair visited, we obtain both the discounted Monte Carlo return and the estimated Q value using $Q_\phi$, and then compute the difference to obtain an estimate of the bias for that state-action pair. We then calculate the average and std of these bias values. The average gives us an idea of whether $Q_\phi$ is in general overestimating or underestimating, and the std measures how uniform the bias is across different state-action pairs. We argue that the std is just as important as the average of the bias. As discussed in Van Hasselt et al. (2016), a uniform bias is not necessarily harmful as it does not change the action selection. Thus near-uniform bias can be preferable to a highly non-uniform bias with a small average value. Although average bias has been analyzed in several previous works (Van Hasselt et al., 2016; Fujimoto et al., 2018; Anschel et al., 2017), the std does not seem to have received much attention.

Since the MC return values can change significantly throughout training, to make comparisons more meaningful, we define the normalized bias of the estimate $Q_\phi(s, a)$ to be $(Q_\phi(s, a) - Q^\pi(s, a))/|E_{\bar{s}, \bar{a} \sim \pi}[Q^\pi(\bar{s}, \bar{a})]|$, which is simply the bias divided by the absolute value of the expected discounted MC return for state-action pairs sampled from the current policy. We focus on the normalized bias in our analysis since it helps show how large the bias is, compared to the scale of the current MC return.

In this and the subsequent section, we compare REDQ with several algorithms and variants. We emphasize that all of the algorithms and variants use the same code base as used in the REDQ experiments (including using SAC as the underlying off-policy algorithm). The only difference is how the targets are calculated in lines 7 and 8 of Algorithm 1.

We first compare REDQ with two natural algorithms, which we call SAC-20 and ensemble averaging (AVG). SAC-20 is SAC but with $G$ increased from 1 (as in standard SAC) to 20. For AVG, we use an ensemble of Q functions, and when computing the Q target, we take the average of all Q values without any in-target minimization. In these comparisons, all three algorithms use a UTD of $G = 20$. In the later ablation section we also have a detailed discussion on experimental results with Maxmin, and explain why it does not work well for the MuJoCo benchmark when using a large ensemble.

Figure 2 presents the results for Ant; the results for the other three environments are consistent with those for Ant and are shown in the Appendix. For each experiment we use 5 random seeds. We first note REDQ learns significantly faster than both SAC-20 and AVG. Strikingly, relative to the other two algorithms, REDQ has a very low normalized std of bias for most of training, indicating the bias across different in-distribution state-action pairs is about the same. Furthermore, throughout most of training, REDQ has a small and near-constant under-estimation bias. The shaded areas for mean and std of bias are also smaller, indicating that REDQ is robust to random initial conditions.

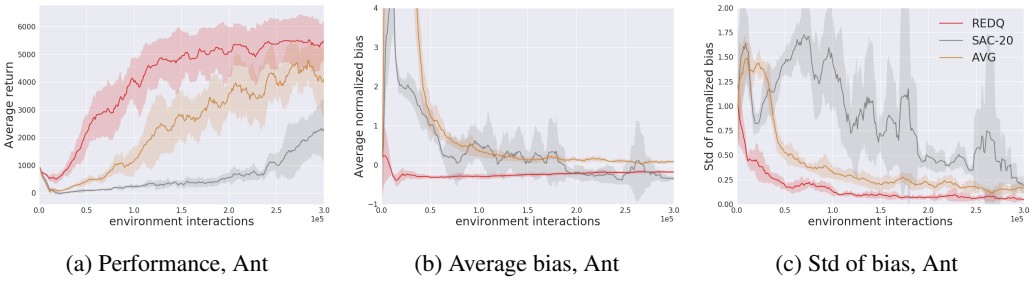

|  (a) Performance, Ant  |  (b) Average bias, Ant  |  (c) Std of bias, Ant  |

Figure 2: Performance, mean and std of normalized Q bias for REDQ, AVG, and SAC for Ant. Figures for the other three environments have similar trends and are shown in the Appendix.

SAC with a UTD ratio of 20 performs poorly for the most challenging environments Ant and Humanoid. For SAC-20, the high UTD ratio leads to an average bias that fluctuates during training. We also see a high normalized std of bias, indicating that the bias is highly non-uniform, which can be detrimental. The bias values also have large variance across random initial seeds, as indicated by the large shaded area, showing that the bias in SAC-20 is sensitive to initial conditions. Comparing AVG and SAC-20, we see AVG performs significantly better than SAC-20 in Ant and Humanoid. This can be explained again by the bias: due to ensemble averaging, AVG can achieve a lower std of bias; and when it does, its performance improves significantly faster than SAC-20.

REDQ has two critical components that allow it to maintain stable and near-uniform bias under high UTD ratios: an ensemble and in-target minimization. AVG and SAC-20 each has one of these components but neither has both. Thus the success of REDQ is largely due to a careful integration of both of these critical components. Additionally, as shown in the ablation study, the *random* selection of Q functions in the in-target minimization can give REDQ a further performance boost.

### 3.1 THEORETICAL ANALYSIS

We now characterize the relation between the estimation error, the in-target minimization parameter $M$ and the size of the ensemble $N$. We use the theoretical framework introduced in Thrun & Schwartz (1993) and extended in Lan et al. (2020). We do this for the tabular version of REDQ, for which the target for $Q^i(s, a)$ for each $i = 1, \ldots, N$ is:

$$r + \gamma \max_{a' \in \mathcal{A}} \min_{j \in \mathcal{M}} Q^j(s', a') \tag{1}$$

where $\mathcal{A}$ is the finite action space, $(s, a, r, s')$ is a transition, and $\mathcal{M}$ is again a uniformly random subset from $\{1, \ldots, N\}$ with $|\mathcal{M}| = M$. The complete pseudocode for tabular REDQ is provided in the Appendix.

Let $Q^i(s, a) - Q^\pi(s, a)$ be the *pre-update* estimation bias for the $i$th Q-function, where $Q^\pi(s, a)$ is once again the ground-truth Q-value for the current policy $\pi$. We are interested in how the bias changes after an update, and how this change is effected by $M$ and $N$. Similar to Thrun & Schwartz (1993) and Lan et al. (2020), define the *post-update* estimation bias as the difference between the target (1) and the target when using the ground-truth:

$$Z_{M,N} \triangleq r + \gamma \max_{a' \in \mathcal{A}} \min_{j \in \mathcal{M}} Q^j(s', a') - (r + \gamma \max_{a' \in \mathcal{A}} Q^\pi(s', a'))$$
$$= \gamma(\max_{a' \in \mathcal{A}} \min_{j \in \mathcal{M}} Q^j(s', a') - \max_{a' \in \mathcal{A}} Q^\pi(s', a'))$$

Here we write $Z_{M,N}$ to emphasize its dependence on both $M$ and $N$. Following Thrun & Schwartz (1993) and Lan et al. (2020), fix $s$ and assume each $Q^i(s, a)$ has a random approximation error $e_{sa}^i$:

$$Q^i(s, a) = Q^\pi(s, a) + e_{sa}^i$$

where for each fixed $s$, $\{e_{sa}^i\}$ are zero-mean independent random variables such that $\{e_{sa}^i\}$ are identically distributed across $i$ for each fixed $(s, a)$ pair. Note that due to the zero-mean assumption, the expected pre-update estimation bias is $\mathbb{E}[Q^i(s, a) - Q^\pi(s, a)] = 0$. Thus if $\mathbb{E}[Z_{M,N}] > 0$, then the expected post-update bias is positive and there is a tendency for over-estimation accumulation; and if $\mathbb{E}[Z_{M,N}] < 0$, then there is a tendency for under-estimation accumulation.

**Theorem 1.**       *1. For any fixed $M$, $\mathbb{E}[Z_{M,N}]$ does not depend on $N$.*

2. *$\mathbb{E}[Z_{1,N}] \geq 0$ for all $N \geq 1$.*

3. *$\mathbb{E}[Z_{M+1,N}] \leq \mathbb{E}[Z_{M,N}]$ for any $M < N$.*

4. *Suppose that $e_{sa}^i \leq c$ for some $c > 0$ for all $s$, $a$ and $i$. Then there exists an $M$ such that for all $N \geq M$, $\mathbb{E}[Z_{M,N}] < 0$.*

Points 2-4 of the Theorem indicate that we can control the expected post-update bias $\mathbb{E}[Z_{M,N}]$, bringing it from above zero (over estimation) to under zero (under estimation) by increasing $M$. Moreover, from the first point, the expected bias only depends on $M$, which implies that increasing $N$ can reduce the variance of the ensemble average but does not change the expected post-update bias. Thus, we can control the post-update bias with $M$ and separately control the variance of the average of the ensemble with $N$. Note that this is not possible for Maxmin Q-learning, for which $M = N$, and thus increasing the ensemble size $N$ will also decrease the post-update bias, potentially making it more negative, which can be detrimental. Note that this also cannot be done for standard ensemble averaging, for which there is no in-target minimization parameter $M$.

Note we make very weak assumptions on the distribution of the error term. Thrun & Schwartz (1993) and Lan et al. (2020) make a strong assumption, namely, the error term is uniformly distributed. Because our assumption is much weaker, our proof methodology in the Appendix is very different.

We also consider a variant of REDQ where instead of choosing a random set of size $M$ in the target, we calculate the target by taking the expected value over all possible subsets of size $M$. In this case, the target in the tabular version becomes

$$Y_{M,N} = r(s, a) + \gamma \frac{1}{\binom{N}{M}} \sum_{\substack{B \subset \mathcal{N} \\ |B| = M}} \max_{a'} \min_{j \in B} Q^j(s', a')$$

We write the target here as $Y_{M,N}$ to emphasize its dependence on both $M$ and $N$. We refer to this variant of REDQ as "Weighted" since we can efficiently calculate the target as a weighted sum of a re-ordering of the $N$ Q-functions, as described in the Appendix.

The following theorem shows that the variance of this target goes to zero as $N \to \infty$. We note, however, that in practice, some variance in the target may be beneficial in reducing overfitting or help exploration. We can retain some variance by keeping $N$ finite or using the unweighted REDQ scheme.

Let $v_M := \text{Var}(\max_{a'} \min_{j \in B} Q^j(s', a'))$ for any subset $B \subset \mathcal{N}$ where $|B| = M$. (It is easily seen that $v_M$ only depends on $M$ and not only the specific elements of $B$.)

**Theorem 2.**

$$\text{Var}(Y_{M,N}) \leq G_M(N)$$

*for some function $G_M(N)$ satisfying*

$$\lim_{N \to \infty} \frac{G_M(N)}{M^2 v_M / N} = 1$$

*Consequently,*

$$\lim_{N \to \infty} \text{Var}(Y_{M,N}) = 0$$

Also in the Appendix we show that the tabular version of REDQ convergences to the optimal Q function with probability one.

## 4 REDQ VARIANTS AND ABLATIONS

In this section, we use ablations to provide further insight into REDQ. We focus on the Ant environment. We first look at how the ensemble size $N$ affects REDQ. The top row in Figure 3 shows REDQ with $N = 2, 3, 5, 10, 15$. We can see that when we increase the ensemble size, we generally get a more stable average bias, a lower std of bias, and stronger performance. The result shows that even a small ensemble (e.g., $N = 5$) can greatly help in stabilizing bias accumulation when training under high UTD.

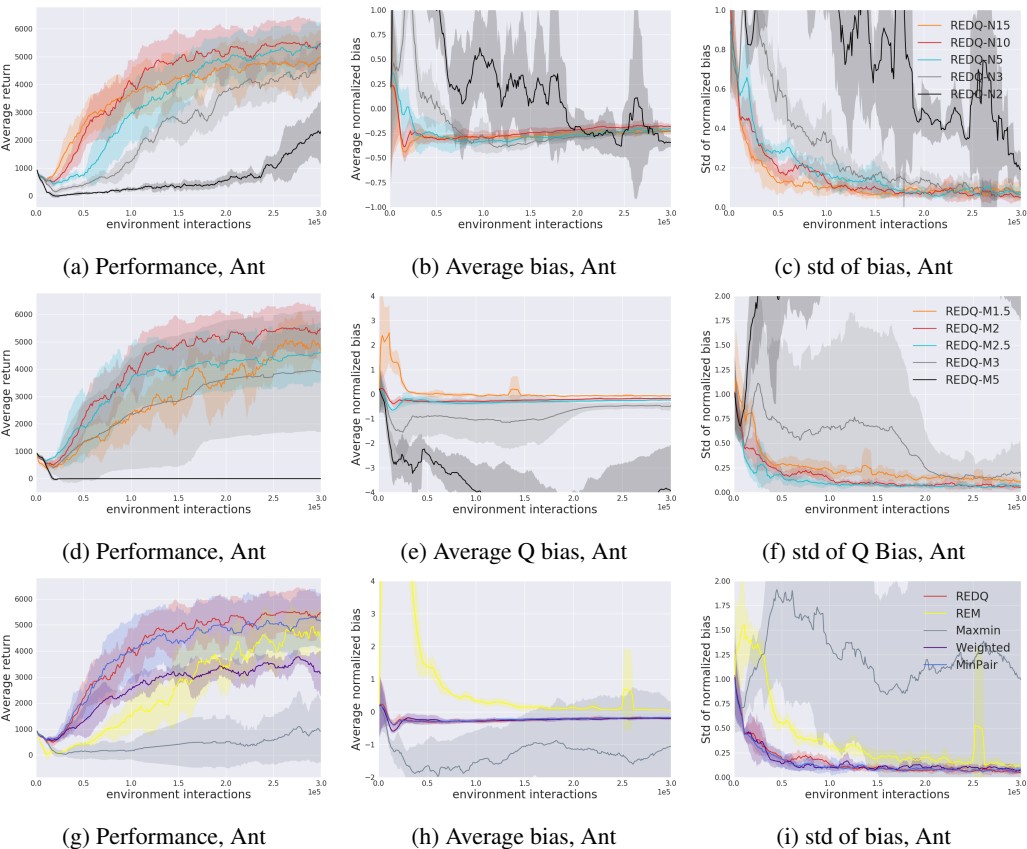

Figure 3: REDQ ablation results for Ant. The top row shows the effect of the ensemble size $N$. The middle row shows the effect of the in-target minimization parameter $M$. The bottom row compares REDQ to several variants.

The middle row of Figure 3 shows how $M$, the in-target minimization parameter, can affect performance. When $M$ is not an integer, e.g., $M = 1.5$, for each update, with probability 0.5 only one

randomly-chosen Q function is used in the target, and with probability 0.5, two randomly-chosen functions are used. Similarly, for $M = 2.5$, for each update either two or three Q functions are used. Consistent with the theoretical result in Theorem 1, by increasing $M$ we lower the average bias. When $M$ gets too large, the Q estimate becomes too conservative and the large negative bias makes learning difficult.

$M = 2$, which has the overall best performance, strikes a good balance between average bias (small underestimation during most of training) and std of the bias (consistently small).

The bottom row of Figure 3 shows the results for different target computation methods. The Maxmin curve in the figures is a variant based on Maxmin Q-learning, where the min of *all* the Q networks in the ensemble is taken to compute the Q target. As the ensemble size increases, Maxmin Q-learning shifts from overestimation to underestimation (Lan et al., 2020); Figure 3 shows Maxmin with $N = 3$ instead of $N = 10$, since a large $N$ value will cause even more divergence of the Q values. When varying the ensemble size of Maxmin, we see the same problem as shown in the middle row of Figure 3. When we increase the ensemble size to be larger than 3, Maxmin starts to reduce the bias so much that we get a highly negative Q bias, which accumulates quickly, leading to instability in the Q networks and poor performance. In the Maxmin paper, it was mainly tested on Atari environments with a small finite action space, in which case it provides good performance. Our results show that when using environments with high-dimensional continuous action spaces, such as MuJoCo, the rapid accumulation of (negative) bias becomes a problem. This result parallels some recent research in offline (i.e., batch) DRL. In Agarwal et al. (2020), it is shown that with small finite action spaces, naive offline training with deep Q-networks (DQN) only slightly reduces performance. However, continuous action Q-learning based methods such as Deep Deterministic Policy Gradient and SAC suffer much more from Q bias accumulation compared to discrete action methods. Recent work shows that offline training with these methods often lead to poor performance, and can even entirely diverge (Fujimoto et al., 2019; Kumar et al., 2019).

Random ensemble mixture (REM) is a method originally proposed to boost performance of DQN in the discrete-action setting. REM uses the random convex combination of Q values to compute the target: it is similar to ensemble average (AVG), but with more randomization (Agarwal et al., 2020).

For Weighted, the target is computed as the expectation of all the REDQ targets, where the expectation is taken over all $N$-choose-2 pairs of Q-functions. This leads to a formula that is a weighted sum of the ordered Q-functions, where the ordering is from the lowest to the highest Q value in the ensemble, as described in the Appendix. Our baseline REDQ in Algorithm 1 can be considered as a random-sample version of Weighted. For the MinPair REDQ variant, we divide the 10 Q networks into 5 fixed pairs, and during an update we sample a pair of Q networks from these 5 fixed pairs.

From Figure 3 we see that REDQ and MinPair are the best and their performance is similar. For Ant, the performance of Weighted is much lower than REDQ. However, as shown in the Appendix, Weighted and REDQ have similar performance for the other three environments. The randomization might help alleviate overfitting in the early stage, or improve exploration. REM has performance similar to AVG, studied in Section 3. In terms of the Q bias, REM has a positive average bias, while REDQ, MinPair, and Weighted all have a small negative average bias. Overall these results indicate that the REDQ algorithm is robust across different mechanisms for choosing the functions, and that randomly choosing the Q functions can sometimes boost performance. Additional results and discussions are provided in the appendix.

### 4.1 IMPROVING REDQ WITH AUXILIARY FEATURE LEARNING

We now investigate whether we can further improve the performance of REDQ by incorporating better representation learning? Ota et al. (2020) recently proposed the online feature extractor network (OFENet), which learns representation vectors from environment data, and provides them to the agent as additional input, giving significant performance improvement.

Is it possible to further improve the performance of REDQ with OFENet? We trained an OFENet together with REDQ to provide extra input, giving the algorithm REDQ-OFE. We found that OFENet did not help much for Hopper and Walker2d, which may be because REDQ already learns very fast, leaving little room for improvement. But as shown in Figure 4, online feature extraction can further improve REDQ performance for the more challenging environments Ant and Humanoid.

REDQ-OFE achieves 7x the sample efficiency of SAC to reach 5000 on Ant and Humanoid, and outperforms MBPO with 3.12x and 1.26x the performance of MBPO at 150K and 300K data, respectively. A more detailed performance comparison table can be found in the Appendix.

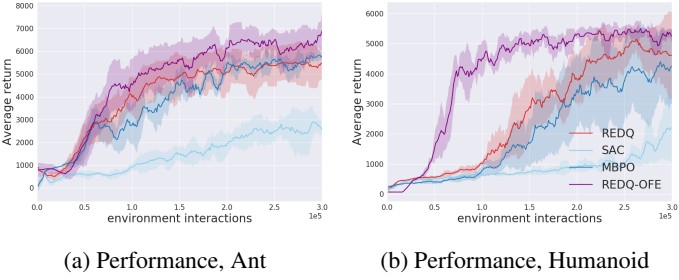

(a) Performance, Ant         (b) Performance, Humanoid

Figure 4: Performance of REDQ, REDQ with OFE, and SAC.

## 5 RELATED WORK

It has long been recognized that maximization bias in Q-learning can significantly impede learning. Thrun & Schwartz (1993) first highlighted the existence of maximization bias. Van Hasselt (2010) proposed Double Q-Learning to address maximization bias for the tabular case, and showed that in general it leads to an under-estimation bias. Van Hasselt et al. (2016) showed that adding Double Q-learning to deep Q networks (DQN) (Mnih et al., 2013; 2015) gives a major performance boost for the Atari games benchmark. For continuous-action spaces, Fujimoto et al. (2018) introduced clipped-double Q-learning (CDQ), which further reduces maximization bias and brings significant improvements over the deep deterministic policy gradient (DDPG) algorithm (Lillicrap et al., 2015). CDQ was later combined with entropy maximization in SAC to achieve even stronger performance (Haarnoja et al., 2018a;b). Other bias reduction techniques include using bias-correction terms (Lee et al., 2013), using weighted Q estimates (Zhang et al., 2017; Li & Hou, 2019), penalizing deterministic policies at early stage of training (Fox et al., 2015), using multi-step methods (Meng et al., 2020), performing weighted Bellman updates to mitigate error propagation (Lee et al., 2020), and truncating sampled Q estimates with distributional networks (Kuznetsov et al., 2020).

It has also long been recognized that using ensembles can improve the performance of DRL algorithms (Faußer & Schwenker, 2015; Osband et al., 2016). For Q-learning based methods, Anschel et al. (2017) use the average of multiple Q estimates to reduce variance. Agarwal et al. (2020) introduced Random Ensemble Mixture (REM), which enforces optimal Bellman consistency on random convex combinations of multiple Q estimates. Lan et al. (2020) introduced Maxmin Q-learning, as discussed in Sections 2-4. Although in discrete action domains it has been found that fine-tuning DQN variants, including the UTD, can boost performance, little experimental or theoretical analysis is given to explain how this improvement is obtained (Kielak, 2020; van Hasselt et al., 2019).

To address some of the critical issues in model-based learning (Langlois et al., 2019), recent methods such as MBPO combine a model ensemble with a carefully controlled rollout horizon to obtain better performance (Janner et al., 2019; Buckman et al., 2018). These model-based methods can also be enhanced with advanced sampling (Zhang et al., 2020), bidirectional models (Lai et al., 2020), or backprop through the model (Clavera et al., 2020), and be analyzed through new theoretical frameworks (Rajeswaran et al., 2020; Dong et al., 2020).

## 6 CONCLUSION

The contributions of this paper are as follows. (1) We propose a simple model-free algorithm that attains sample efficiency that is as good as or better than state-of-the-art model-based algorithms for the MuJoCo benchmark. This result indicates that, at least for the MuJoCo benchmark, models may not be necessary for achieving high sample efficiency. (2) Using carefully designed experiments, we explain why REDQ succeeds when other model-free algorithms with high UTD ratios fail. (3) Finally, we combine REDQ with OFE, and show that REDQ-OFE can learn extremely fast for the challenging environments Ant and Humanoid.

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

# A    THEORETICAL RESULTS

## A.1    TABULAR VERSION OF REDQ

In the tabular algorithm below, for clarity we use $G = 1$.

---
**Algorithm 2** Tabular REDQ
---
1: Initialize $\{Q^i(s,a), s \in \mathcal{S}, a \in \mathcal{A}\}_{i=1}^N$,
2: **repeat**
3:      Choose $a \in \mathcal{A}$ based on $\{Q^i(s,a)\}_{i=1}^N$, observe $r, s'$
4:      Randomly choose a subset $\mathcal{M}$ of size $M$ from $\{1, .., N\}$
5:      $y = r + \gamma \max_{a' \in \mathcal{A}} \min_{j \in \mathcal{M}} Q^j(s', a')$
6:      **for** $i = 1, \ldots, N$ **do**:
7:          $Q^i(s,a) \leftarrow Q^i(s,a) + \alpha(y - Q^i(s,a))$
8:      $s \leftarrow s'$
9: **until** end

---

Alternatively in Algorithm 2, at each iteration we could just update one of the $Q^i$ functions.

## A.2    PROOF OF THEOREM 1

We first prove the following lemma:

**Lemma 1.** *Let $X_1, X_2, \ldots$ be an infinite sequence of i.i.d. random variables. Let $F(x)$ be the cdf of $X_m$ and let $\tau = \inf\{x : F(x) > 0\}$. Also let $Y_m = \min\{X_1, X_2, \ldots, X_m\}$. Then $Y_1, Y_2, \ldots$ converges to $\tau$ almost surely.*

**Proof:**

Let $F_m(x)$ be the cdf of $Y_m$. Since $X_1, ..., X_m$ are independent,

$$F_m(x) = 1 - [1 - F(x)]^m$$

For $x < \tau$, $F_m(x) = 0$ since $F(x) = 0$. For $x > \tau$, $F_m(x) \xrightarrow{m \to \infty} 1$. Therefore, $Y_m$ weakly converges to $\tau$.

Moreover, for each $\omega \in \Omega$, $\{Y_m(\omega)\}$ is a decreasing sequence. So $\{Y_m(\omega)\}$ either converges to a real number or $-\infty$. Therefore $Y_m \to Y$ almost surely for some random variable $Y$. Combined with the result that $Y_m \xrightarrow{d} \tau$, we can conclude that

$$Y_m \xrightarrow{a.s.} \tau$$

**Proof of Theorem 1:**

1. Let $B_1$, $B_2$ be two subsets of $\mathcal{N} = \{1, ..., N\}$ of size $M$. First of all, since $\{Q^j(s,a)\}_{j=1}^N$ are i.i.d for any $a \in \mathcal{A}$, $\min_{j \in B_1} Q^j(s,a)$ and $\min_{j \in B_2} Q^j(s,a)$ are identically distributed. Furthermore, since $Q^j(s,a)$ are independent for all $a \in \mathcal{A}$ and $1 \le j \le N$, $\left\{ \min_{j \in B} Q^j(s,a) \right\}_{a \in \mathcal{A}}$ are independent for any $B \subset \mathcal{N}$. Denote the distribution function of $\max_a \min_{j \in B_1} Q^j(s,a)$ as $F_1(x)$ and the distribution function of $\max_a \min_{j \in B_2} Q^j(s,a)$ as $F_2(x)$. Then for any $x \in \mathbb{R}$,

$$F_1(x) = \mathbb{P}\left( \max_a \min_{j \in B_1} Q^j(s,a) \le x \right) = \mathbb{P}\left( \cap_{a \in \mathcal{A}} \left\{ \min_{i \in B_1} Q^j(s,a) \le x \right\} \right)$$

$$= \prod_{a \in \mathcal{A}} \mathbb{P}\left( \min_{j \in B_1} Q^j(s,a) \le x \right) = \prod_{a \in \mathcal{A}} \mathbb{P}\left( \min_{j \in B_2} Q^j(s,a) \le x \right)$$

$$= \mathbb{P}\left( \max_a \min_{j \in B_2} Q^j(s,a) \le x \right) = F_2(x)$$

Therefore, we have proved that $\max_a \min_{j \in B_1} Q^j(s, a)$ and $\max_a \min_{j \in B_2} Q^j(s, a)$ are identically distributed. Then

$$\mathbb{E}[Z_{M,N}] = \gamma \mathbb{E}\big[(\max_a \min_{j \in \mathcal{M}} Q^j(s, a) - \max_a Q^\pi(s, a))\big]$$

$$= \gamma \mathbb{E}\big[\frac{1}{\binom{N}{M}} \sum_{\substack{B \subset \mathcal{N} \\ |B| = M}} \max_a \min_{j \in B} Q^j(s, a)\big] - \gamma \max_a Q^\pi(s, a)$$

$$= \gamma \bigg( \mathbb{E}\big[\max_a \min_{1 \leq j \leq M} Q^j(s, a)\big] - \max_a Q^\pi(s, a) \bigg)$$

which does not depend on $N$.

2. It follows from 1 that

$$\mathbb{E}[Z_{1,N}] = \gamma \bigg( \mathbb{E}\big[\max_a Q^1(s, a)\big] - \max_a Q^\pi(s, a) \bigg)$$

Since $\max_a Q^1(s, a) \geq Q^1(s, a')$ for all $a' \in \mathcal{A}$, we have

$$\mathbb{E}\big[\max_a Q^1(s, a)\big] \geq \mathbb{E}\big[Q^1(s, a')\big]$$

for all $a' \in \mathcal{A}$. Consequently,

$$\mathbb{E}\big[\max_a Q^1(s, a)\big] \geq \max_a \mathbb{E}\big[Q^1(s, a)\big] = \max_a Q^\pi(s, a)$$

Hence

$$\mathbb{E}[Z_{1,N}] = \gamma \bigg( \mathbb{E}\big[\max_a Q^1(s, a)\big] - \max_a Q^\pi(s, a) \bigg) \geq 0$$

3. Since $\max_a \min_{1 \leq j \leq M} Q^j(s, a) \geq \max_a \min_{1 \leq j \leq M+1} Q^j(s, a)$,

$$\mathbb{E}[Z_{M,N}] = \gamma \bigg( \mathbb{E}\big[\max_a \min_{1 \leq j \leq M} Q^j(s, a)\big] - \max_a Q^\pi(s, a) \bigg)$$

$$\geq \gamma \bigg( \mathbb{E}\big[\max_a \min_{1 \leq j \leq M+1} Q^j(s, a)\big] - \max_a Q^\pi(s, a) \bigg) = \mathbb{E}[Z_{M+1,N}]$$

4. Let $F_a(x)$ be the cdf of $Q^j(s, a)$ and let $\tau_a = \inf\{x : F_a(x) > 0\}$. Here we assume the approximation error $e^i_{sa}$ is non-trivial, which implies $\tau_a < Q^\pi(s, a)$. Note that $\tau_a$ can be equal to $-\infty$. Let

$$Y^M_a = \min_{1 \leq i \leq M} Q^j(s, a)$$

From Lemma 1 we have $Y^M_a$ converges to $\tau_a$ almost surely for each $a$. Because the action space is finite, it therefore follows that

$$Y^M = \max_a Y^M_a$$

converges almost surely to $\tau = \max_a \tau_a$. Furthermore, for each $a$ we have

$$Y^M_a = \min_{1 \leq j \leq M} Q^j(s, a) \geq \min_{1 \leq j \leq M+1} Q^j(s, a) = Y^{M+1}_a$$

from which it follows that $Y^M \geq Y^{M+1}$. Thus $\{Y^M\}$ is a monotonically decreasing sequence. We also note that due to the assumption $e^i_{sa} \leq c$ for all $a$ and $i$, and because $Q^\pi(s, a)$ is finite for all $s$ and $a$, it follows that $Y^M \leq d$ for all $M$ for a finite $d$. Thus $\{Y^M\}$ is a bounded-above, monotonically-decreasing sequence of random variables which converges almost surely to $\tau$. We can therefore apply the monotone convergence theorem, giving

$$\mathbb{E}[Z_{M,N}] = \gamma \bigg( \mathbb{E}\big[\max_a \min_{1 \leq j \leq M} Q^j(s, a)\big] - \max_a Q^\pi(s, a) \bigg)$$

$$= \gamma \bigg( \mathbb{E}[\max_a Y^M_a] - \max_a Q^\pi(s, a) \bigg) \xrightarrow{M \to \infty} \gamma \bigg( \max_a \tau_a - \max_a Q^\pi(s, a) \bigg) < 0,$$

where the last inequality follows from $\tau_a < Q^\pi(s, a)$ for all actions $a$.

## A.3 PROOF OF THEOREM 2

For convenience, define $Y_B = \max_{a'} \min_{j \in B} Q^j(s', a')$. Suppose $N > 2M$.

$$
\begin{aligned}
\mathrm{Var}(Y_{M,N}) &= \frac{\gamma^2}{\binom{N}{M}^2} \mathrm{Var}\Big( \sum_{\substack{B \subset \mathcal{N} \\ |B|=M}} Y_B \Big) \\
&= \frac{\gamma^2 (M!)^2}{\left( \Pi_{i=0}^{M-1}(N-i) \right)^2} \Big[ \sum_{B \subset \mathcal{N}} \mathrm{Var}(Y_B) + 2 \cdot \sum_{\substack{B_1, B_2 \subset \mathcal{N} \\ B_1 \neq B_2}} \mathrm{Cov}(Y_{B_1}, Y_{B_2}) \Big]
\end{aligned}
$$

Let $A = \sum_{\substack{B_1, B_2 \subset \mathcal{N} \\ B_1 \neq B_2}} \mathrm{Cov}(Y_{B_1}, Y_{B_2})$, which consists of

$$
\begin{aligned}
\binom{\binom{N}{M}}{2} &= \frac{1}{2} \cdot \frac{N!}{(N-M)!M!} \cdot \Big( \frac{N!}{(N-M)!M!} - 1 \Big) \\
&= \frac{1}{2(M!)^2} \cdot \Pi_{i=0}^{M-1}(N-i)^2 - \frac{N!}{2 \cdot M!(N-M)!}
\end{aligned}
$$

terms. $\binom{\binom{N}{M}}{2}$ can be seen as a polynomial function of $N$ with degree $2M$. The coefficient for the term $N^{2M}$ is $\frac{1}{2(M!)^2}$. The coefficient for the term $N^{2M-1}$ is $\frac{1}{2(M!)^2} \cdot (-2 \sum_{i=0}^{M-1} i)$.

Note that $Y_{B_1}$ and $Y_{B_2}$ are independent if $B_1 \cap B_2 = \varnothing$. The total number of different pairs $(B_1, B_2)$ such that $B_1 \cap B_2 = \varnothing$ is

$$
\binom{N}{2M} \cdot \binom{2M}{M} \cdot \frac{1}{2} = \frac{1}{2(M!)^2} \cdot \frac{N!}{(N-2M)!} = \frac{1}{2(M!)^2} \cdot \Pi_{i=0}^{2M-1}(N-i)
$$

This is again a polynomial function of $N$ with degree $2M$. The coefficient of the term $N^{2M}$ is $\frac{1}{2(M!)^2}$. The coefficient of the term $N^{2M-1}$ is $\frac{1}{2(M!)^2} \cdot (-\sum_{i=0}^{2M-1} i)$. So the number of non-zero terms in $A$ is at most

$$
\begin{aligned}
&\frac{1}{2(M!)^2} \cdot \Pi_{i=0}^{M-1}(N-i)^2 - \frac{N!}{2 \cdot M!(N-M)!} - \frac{1}{2(M!)^2} \cdot \Pi_{i=0}^{2M-1}(N-i) \\
&= \frac{M^2}{2(M!)^2} \cdot N^{2M-1} + O(N^{2M-2})
\end{aligned}
$$

Moreover, by Cauchy-Schwarz inequality, for any $B_1, B_2 \subset \mathcal{N}$

$$
\mathrm{Cov}(Y_{B_1}, Y_{B_2}) \leq \sqrt{\mathrm{Var}(Y_{B_1}) \cdot \mathrm{Var}(Y_{B_2})} = v_M
$$

Therefore,

$$
A \leq \big[ \frac{M^2}{2(M!)^2} \cdot N^{2M-1} + O(N^{2M-2}) \big] v_M
$$

which implies

$$
\begin{aligned}
\mathrm{Var}(Y_{M,N}) &= \frac{\gamma^2 (M!)^2}{\left( \Pi_{i=0}^{M-1}(N-i) \right)^2} \Big[ \sum_{B \subset \mathcal{N}} \mathrm{Var}(Y_B) + 2 \cdot \sum_{\substack{B_1, B_2 \subset \mathcal{N} \\ B_1 \neq B_2}} \mathrm{Cov}(Y_{B_1}, Y_{B_2}) \Big] \\
&= \frac{\gamma^2 (M!)^2}{\left( \Pi_{i=0}^{M-1}(N-i) \right)^2} \Big[ \sum_{B \subset \mathcal{N}} \mathrm{Var}(Y_B) + 2A \Big] \\
&\leq \gamma^2 \big[ M^2 \cdot \frac{N^{2M-1}}{\Pi_{i=0}^{M-1}(N-i)^2} + O(\frac{1}{N^2}) \big] v_M \xrightarrow{N \to \infty} 0
\end{aligned}
$$

Moreover,

$$
\lim_{N \to \infty} \frac{M^2 v_M / N}{\big[ M^2 \cdot \frac{N^{2M-1}}{\Pi_{i=0}^{M-1}(N-i)^2} + O(\frac{1}{N^2}) \big] v_M} = \lim_{N \to \infty} \frac{\Pi_{i=0}^{M-1}(N-i)^2}{N^{2M}} = 1
$$

A.4    PROOF OF CONVERGENCE OF TABULAR REDQ

Assuming that the step size satisfies the standard Robbins-Monro conditions, it is easily seen that the tabular version of REDQ converges with probability 1 to the optimal Q function. In fact, for our Weighted scheme, where we take the expectation over all sets of size $M$, the convergence conditions in Lan et al. (2020) are fully satisfied.

For the randomized case, only very minor changes are needed in the proof in Lan et al. (2020). Note that in the case of REDQ, the underlying deterministic target is:

$$F\left(Q^1, Q^2, \ldots, Q^N\right)(s, a) = r(s, a) + \gamma \sum_{s'} p\left(s' \mid s, a\right) \sum_{\substack{B \subset \mathcal{N} \\ |B|=M}} \frac{1}{\binom{N}{M}} \max_{a'} \min_{j \in B} Q^j(s', a')$$

Let $\mathcal{T}$ be the operator that concatenates $N$ identical copies of $F$, so that $\mathcal{T}\colon \mathbb{R}^{S \times A \times N} \to \mathbb{R}^{S \times A \times N}$ where $S$ and $A$ are the cardinalities of the state and action spaces, respectively. It is easy to show that the operator $\mathcal{T}$ is a contraction with the $l_\infty$ norm. The stochastic approximation noise term is given by

$$\omega(s, a) = R - r(s, a) + \gamma \left[ \max_{a'} \min_{j \in B} Q^j(s', a') - \sum_{s'} p\left(s' \mid s, a\right) \sum_{\substack{B \subset \mathcal{N} \\ |B|=M}} \frac{1}{\binom{N}{M}} \max_{a'} \min_{j \in B} Q^j(s', a') \right]$$

It is straightforward to show

$$\mathbb{E}\left[\omega^2(s, a) \mid \mathcal{F}_{\text{past}}\right] \leq \operatorname{Var}\left(R \mid s, a\right) + \max_{1 \leq i \leq N} \max_{s', a'} \left(Q^i(s', a')\right)^2 \tag{2}$$

As in Lan et al. (2020), it follows from the contraction property and (2) that REDQ converges with probability 1 to the optimal Q function (Tsitsiklis, 1994; Bertsekas & Tsitsiklis, 1996).

## B  HYPERPARAMETERS AND IMPLEMENTATION DETAILS

Since MBPO builds on top of a SAC agent, to make our comparisons fair, meaningful, and consistent with previous work, we make all SAC related hyperparameters exactly the same as used in the MBPO paper (Janner et al., 2019). Table 1 gives a list of hyperparameter used in the experiments. For all the REDQ curves reported in the results section, we use a Q network ensemble size $N$ of 10. We use a UTD ratio $G$ of 20 on the four MuJoCo environments, which is the same value that was used in the MBPO paper. Thus most of the hyperparameters are made to be the same as in the MBPO paper to ensure fairness and consistency in comparisons.

For all the algorithms and variants, we also first obtain 5000 data points by randomly sampling actions from the action space without making parameter updates. In our experiments we found that using a high UTD from the very beginning with a very small amount of data can easily lead to complete divergence on SAC-20. Sampling a number of random datapoints at the start of training is also a common technique that has been used in previous model-free as well as model-based works (Haarnoja et al., 2018a; Fujimoto et al., 2018; Janner et al., 2019).

Table 1: REDQ hyperparameters

| Parameter | Value |
|---|---|
| *Shared* | |
| optimizer | Adam (Kingma & Ba, 2014) |
| learning rate | $3 \cdot 10^{-4}$ |
| discount ($\gamma$) | 0.99 |
| target smoothing coefficient ($\rho$) | 0.005 |
| replay buffer size | $10^6$ |
| number of hidden layers for all networks | 2 |
| number of hidden units per layer | 256 |
| mini-batch size | 256 |
| nonlinearity | ReLU |
| random starting data | 5000 |
| *REDQ* | |
| ensemble size $N$ | 10 |
| in-target minimization parameter $M$ | 2 |
| UTD ratio $G$ | 20 |
| *OFENet* | |
| random starting data | 20,000 |
| OFENet number of pretraining updates | 100,000 |
| OFENet UTD ratio | 4 |

For the REDQ-OFE experiments, we implemented a minimal version of the OFENet in the original paper, with no batchnorm layers. We use the recommended hyperparameters as described in the original paper (Ota et al., 2020). Compared to REDQ without OFENet, the main difference is we now first collect 20,000 random data points (which is accounted for in the training curves), and then pre-train the OFENet for 100,000 updates, with the same learning rate and batch size. We then train OFENet together with REDQ agent, and the OFENet uses a UTD ratio of 4. We tried a simple hyperparameter search on Ant with 200,000, 100,000 and 50,000 pre-train updates, and learning rates of 1e-4, 3e-4, 5e-4, and a OFENet UTD of 1, 4 and 20. However, the results are not very different. It is possible that better results can be obtained through a more extensive hyperparameter search or other modifications.

### B.1  EFFICIENT CALCULATION OF TARGET FOR WEIGHTED VERSION OF REDQ

In section 4 we provided experimental results for the Weighted version of REDQ. Recall that in this version, instead of sampling a random subset $\mathcal{M}$ in the target, we average over all subsets $B$ in

$\{1, \ldots, N\}$ of size $M$:

$$y = r + \gamma \frac{1}{\binom{N}{M}} \sum_B \left( \min_{i \in B} Q_{\phi_{\text{targ}, i}}(s', \tilde{a}') \right)$$

In practice, however, we do not need to sum over all $N$ choose $M$ subsets. Instead we can re-order the indices so that

$$Q_{\phi_{\text{targ}, i}}(s', \tilde{a}') \leq Q_{\phi_{\text{targ}, i+1}}(s', \tilde{a}')$$

for $i = 1, \ldots, N - 1$. After the re-ordering, we can use the identity:

$$\frac{1}{\binom{N}{M}} \sum_B \left( \min_{i \in B} Q_{\phi_{\text{targ}, i}}(s', \tilde{a}') \right) = \frac{1}{\binom{N}{M}} \sum_{i=1}^{N-M+1} \binom{N-i}{M-1} Q_{\phi_{\text{targ}, i}}(s', \tilde{a}')$$

## C    SAMPLE EFFICIENCY COMPARISON FOR REDQ, SAC AND MBPO

The sample efficiency claims made in the main paper are based on Table 2 and Table 3. Table 2 shows that compared to naive SAC, REDQ is much more sample efficient. REDQ reaches 3500 on Hopper with 8x sample efficiency, and reaches 5000 for Ant and Humanoid with 5x and 3.7x sample efficiency. After adding OFE, this becomes more than 7x on Ant and Humanoid. If we average all the numbers for the four environments, then REDQ is 5.0x as sample efficient, and 6.4x after including OFE results.

Table 3 compares REDQ to SAC and MBPO. As in the MBPO paper, we train for 125K for Hopper, and 300K for the other three environments (Janner et al., 2019). The numbers in Table 3 show the performance when trained to half and to the full length of the MBPO training limits. When averaging the numbers, we see that REDQ reaches 4.5x and 2.1x the performance of SAC at 150K and 300K. REDQ is also stronger than MBPO, with 1.4x and 1.1x the performance of MBPO at 150K and 300K. If we include the results of REDQ-OFE, then the numbers become 5.5x and 2.3x the SAC performance at 150K and 300K, and 1.8x and 1.2x the MBPO performance at 150 and 300K.

Table 2: Sample efficiency comparison of SAC and REDQ. The numbers show the amount of data collected when the specified performance level is reached. The last two columns show how many times REDQ and REDQ-OFE are more sample efficient than SAC in reaching that performance.

| Score | SAC | REDQ | REDQ-OFE | REDQ faster | REDQ-OFE faster |
|---|---|---|---|---|---|
| Hopper at 3500 | 933K | 116K | - | 8.04 | - |
| Walker2d at 3500 | 440K | 141K | - | 3.12 | - |
| Ant at 5000 | 771K | 153K | 106K | 5.04 | 7.27 |
| Humanoid at 5000 | 945K | 255K | 134K | 3.71 | 7.05 |

Table 3: Performance comparison of REDQ, REDQ-OFE, MBPO and SAC. The numbers show the performance achieved when the specific amount of data is collected. The last two columns show the ratio of REDQ or REDQ-OFE performance compared to SAC and MBPO performance.

| Amount of data | SAC | MBPO | REDQ | REDQ/SAC | REDQ/MBPO |
|---|---|---|---|---|---|
| Hopper at 62K | 594 | 1919 | 3278 | 5.52 | 1.71 |
| Hopper at 125K | 2404 | 3131 | 3517 | 1.46 | 1.12 |
| Walker2d at 150K | 760 | 3308 | 3544 | 4.66 | 1.07 |
| Walker2d at 300K | 2556 | 3537 | 4589 | 1.80 | 1.30 |
| Ant at 150K | 1245 | 4388 | 4803 | 3.86 | 1.09 |
| Ant at 300K | 2485 | 5774 | 5369 | 2.16 | 0.93 |
| Humanoid at 150K | 674 | 1604 | 2641 | 3.92 | 1.65 |
| Humanoid at 300K | 1633 | 4199 | 4674 | 2.86 | 1.11 |
| Amount of data | SAC | MBPO | REDQ-OFE | OFE/SAC | OFE/MBPO |
| Ant at 150K | 1245 | 4388 | 5524 | 4.44 | 1.26 |
| Ant at 300K | 2485 | 5774 | 6578 | 2.65 | 1.14 |
| Humanoid at 150K | 674 | 1604 | 5011 | 7.43 | 3.12 |
| Humanoid at 300K | 1633 | 4199 | 5309 | 3.25 | 1.26 |

## D  NUMBER OF PARAMETERS COMPARISON

Table 4 gives the number of parameters for MBPO, REDQ and REDQ-OFE, for all four environments. As discussed in the main paper, REDQ uses fewer parameters than MBPO for all four environments: between 26% and 70% as many parameters depending on the environment. After adding OFENet, REDQ still uses fewer parameters than MBPO, with 80% and 35% as many parameters on Ant and Humanoid. In particular, it is surprising that REDQ-OFE can achieve a much stronger result on Humanoid with much fewer parameters.

Table 4: Number of parameters in millions. REDQ uses the same network structure and ensemble size for all four environments. The difference in the number of parameters comes from the fact that the environments have very different observation and action dimensions, which will affect the size of the input and output layers of the networks.

| Algorithm | Hopper | Walker2d | Ant | Humanoid |
|---|---|---|---|---|
| MBPO | 1.106M | 1.144M | 1.617M | 7.087M |
| REDQ $N = 10$ | 0.769M | 0.795M | 1.066M | 1.840M |
| REDQ-OFE $N = 10$ | - | - | 1.294M | 2.460M |

# E    ADDITIONAL RESULTS FOR REDQ, SAC-20, AND AVG

Due to lack of space, Figure 2 in Section 3 only compared REDQ with SAC-20 and AVG for the
Ant environment. Figure 5 presents the results for all four environments. We can see that in all
four environments, REDQ has much stronger performance and much lower std of bias compared to
SAC-20 and AVG. Note in terms of average normalized bias, AVG is slightly closer to zero in Ant
compared to REDQ, and SAC-20 is a bit closer to zero in Humanoid compared to REDQ; however,
their std of normalized bias is consistently higher. This shows the importance of having a low std of
the bias in addition to a close-to-zero average bias.

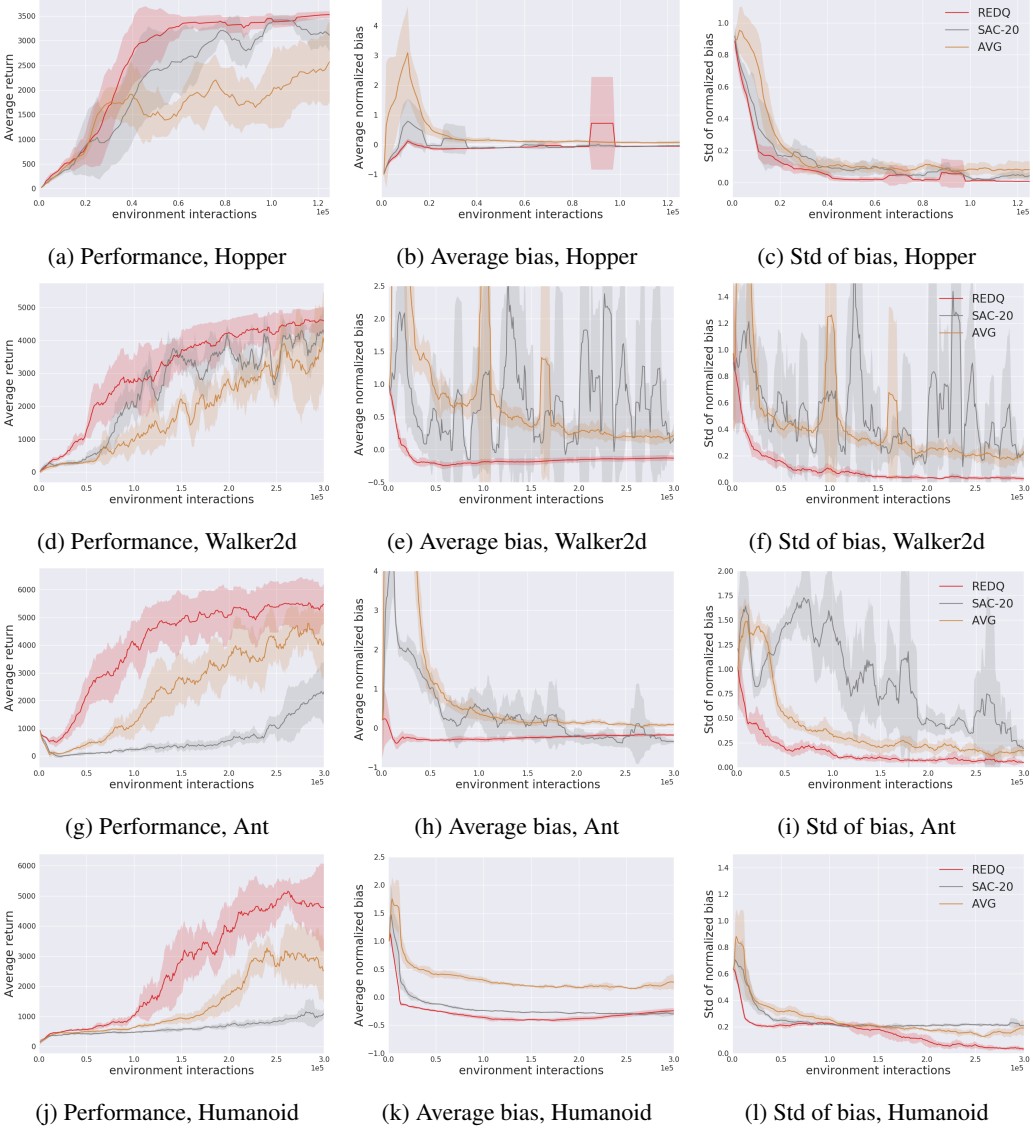

(a) Performance, Hopper    (b) Average bias, Hopper    (c) Std of bias, Hopper

(d) Performance, Walker2d    (e) Average bias, Walker2d    (f) Std of bias, Walker2d

(g) Performance, Ant    (h) Average bias, Ant    (i) Std of bias, Ant

(j) Performance, Humanoid    (k) Average bias, Humanoid    (l) Std of bias, Humanoid

Figure 5: Performance, mean and std of normalized Q bias for REDQ, AVG, and SAC. All three
algorithms have a UTD ratio of 20.

## F    REDQ AND SAC WITH AND WITHOUT POLICY DELAY

Note that in the REDQ pseudocode, the number of policy updates is always one for each data point collected. We set the UTD ratio for the policy update to always be one in order to isolate the effect of additional policy updates from Q updates. Note in this way, REDQ, SAC-20 and SAC-1 all take the same number of policy updates. This helps show that the performance gain mainly comes from the additional Q updates.

Having a lower number of policy updates can also be seen as a delayed policy update, or policy delay, and is a method that has been used in previous works to improve learning stability (Fujimoto et al., 2018). In this section we discuss how delayed policy update, or policy delay, impact the performance of REDQ and SAC (with UTD of 20). Figure 6 compares REDQ and SAC-20 with and without policy delay (NPD for no policy delay). We can see that having the policy delay consistently makes the bias and std of bias lower and more stable, although they have a smaller effect on REDQ than on SAC. Performance-wise SAC always gets a performance boost with policy delay, while REDQ sees improvement in Hopper and Humanoid, and becomes slightly worse in Walker2d and Ant. The results show that policy delay can be important under high UTD when the variance is not properly controlled. However, with enough variance reduction, the effect of policy delay is diminished, and in some cases having more policy update can give better performance.

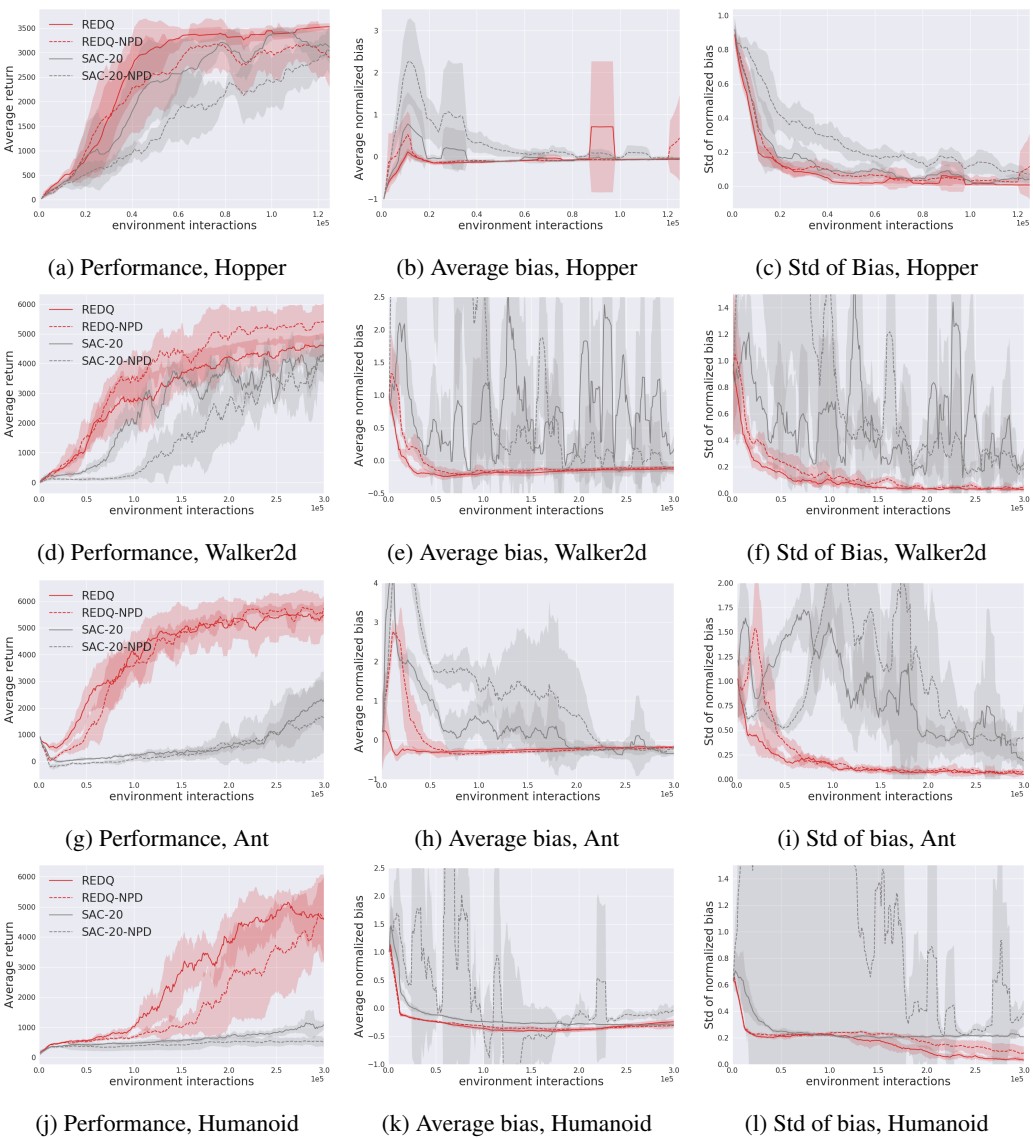

Figure 6: Performance, mean and std of normalized Q bias of REDQ and SAC, with and without policy delay.

# G REDQ AND SAC WITH DIFFERENT UTD RATIOS

How do different UTD ratio values $G$ impact the performance of REDQ and SAC? Figure 7 compares the two algorithms under UTD ratio values of 1, 5, 10 and 20 for the Ant environment. The results show that in the Ant environment, REDQ greatly benefits from larger UTD values, with UTD of 20 giving the best result. For SAC, performance improves slightly for UTD ratios of 5 and 10, but becomes much worse at 20. Looking at the normalized bias and the std of the bias, we see that changing the UTD ratio does not change the values very much for REDQ, while for SAC, we see that as the UTD ratio increases, both the mean and the std of the bias becomes larger and more unstable.

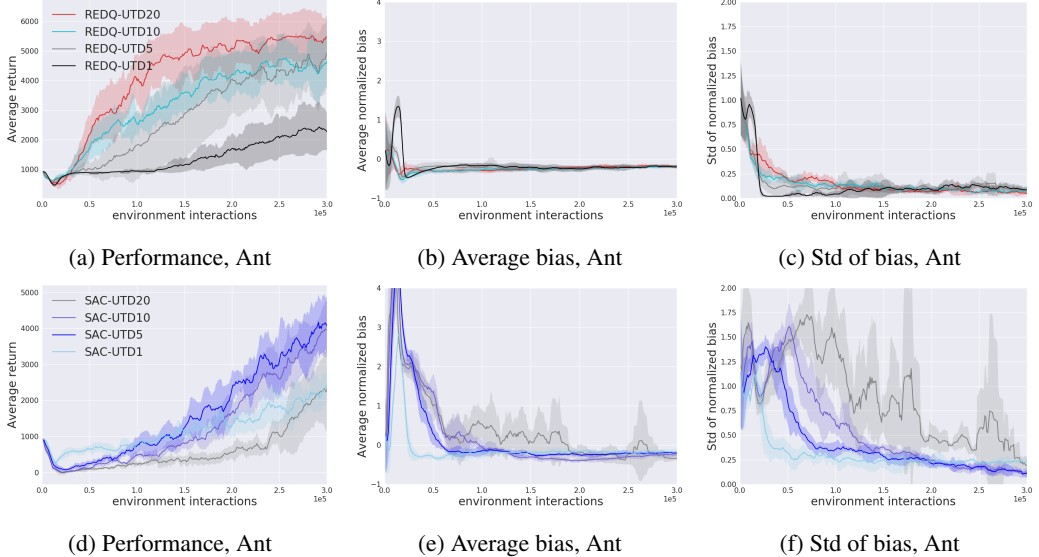

Figure 7: Performance, mean and std of normalized Q bias for REDQ, and SAC, with different UTD ratios, in Ant environment.

## H   ADDITIONAL RESULTS FOR WEIGHTED VARIANT

In this section we provide additional results for the Weighted variant. Figure 8 shows the performance and bias comparison on all four environments. Results show that Weighted and REDQ have similar average bias and std of bias. In terms of performance, Weighed is worse in Ant and Hopper, similar in Humanoid and slightly stronger in Walker2d. Overall REDQ seems to have stronger performance and is more robust. Randomness in the networks might help alleviate overfitting in the early stage, or improve exploration, as shown in previous studies (Osband et al., 2016; Fortunato et al., 2018). This can be important since positive bias in Q learning-based methods can sometimes help exploration. This is commonly referred to as optimistic initial values, or optimism in the face of uncertainty (Sutton & Barto, 2018; Brafman & Tennenholtz, 2002). Thus conservative Q estimates in recent algorithms can lead to the problem of pessimistic underexploration (Ciosek et al., 2019). An interesting future work direction is to study how robust and effective exploration can be achieved without relying on optimistic estimates.

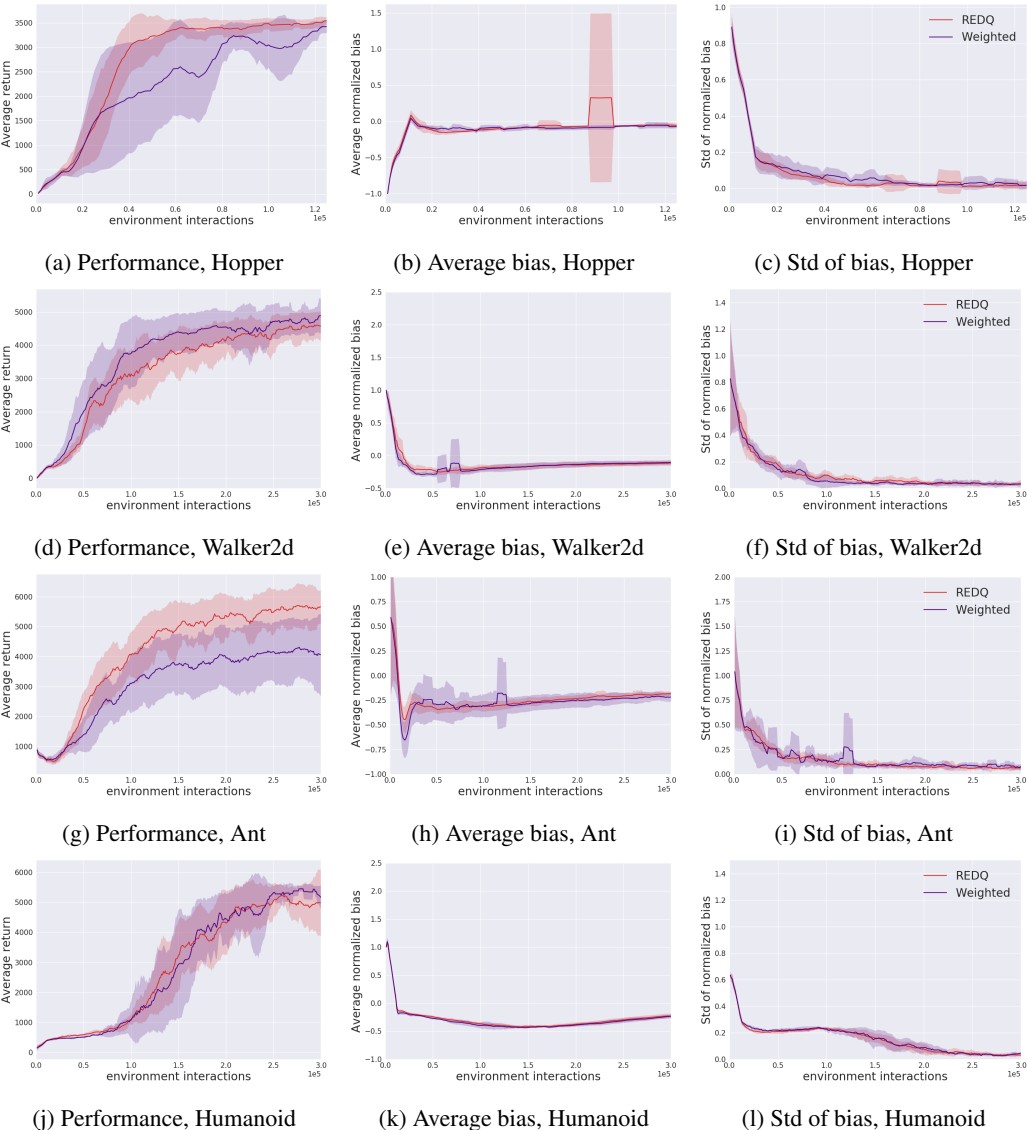

(a) Performance, Hopper   (b) Average bias, Hopper   (c) Std of bias, Hopper

(d) Performance, Walker2d   (e) Average bias, Walker2d   (f) Std of bias, Walker2d

(g) Performance, Ant   (h) Average bias, Ant   (i) Std of bias, Ant

(j) Performance, Humanoid   (k) Average bias, Humanoid   (l) Std of bias, Humanoid

Figure 8: Performance, mean and std of normalized Q bias for REDQ and Weighted.

