# OpenReview forum: "Randomized Ensembled Double Q-Learning: Learning Fast Without a Model"
_ICLR.cc/2021/Conference — ICLR 2021 Poster_

### Official Review · AnonReviewer4 · 2020-10-26
**A variant of double Q-learning, with randomized ensembling**

**Rating:** 7
**Confidence:** 3

**Review:**

This work proposes a modification for double Q-learning, termed as randomized ensembled double Q-learning (REDQ). REDQ maintains $N$ different Q functions, and for each update, the target value is a minimization over $M$ randomly chosen Q functions, where $1 \le M \le N$. In addition, REDQ adopts a high update-to-data ratio to improve the sample efficiency. Empirical results show that the proposed method outperforms state-of-the-art model-based algorithms in certain tasks with continuous action space.

Overall, I think this is a well written paper.

Pros:

- The algorithmic idea of REDQ is quite intuitive and reasonable. In particular, setting $M$ and $N$ separately allows for more flexibility (compared with double Q and other variants): by increasing $M$, one can achieve a smooth transition from over-estimation of value functions to under-estimation (as validated by a simple theoretical analysis). Moreover, this idea is quite general and can be easily plugged into many existing off-policy model-free algorithms.

- The numerical experiments are quite extensive. The results convincingly show that the proposed REDQ algorithm achieves a minor underestimation bias with low standard deviation, leading to better overall performance. In addition, REDQ is also computationally efficient.

Cons:

- The novelty in the algorithmic idea of REDQ is a bit simple and not very significant.

- The theoretical analysis of this work is quite limited, e.g. lacking the (most basic) asymptotic convergence analysis in the tabular case.

A minor comment: it might be better to use "\gg" and "\ll" in latex, instead of ">>" and "<<".

---

> ### Author Response · Authors · 2020-11-18
> **Reply to AnonReviewer4**
>
> Thank you for your positive feedback and detailed review.
>
> Indeed, the algorithmic idea of REDQ is simple and straightforward. However, we argue that applying REDQ to the high UTD ratio setting is very novel and significant. For model-free methods, a UTD of 1 is the most common value for recent algorithms. Our work answers the very important question of “why can’t we exploit the data further with more updates”. Our experiments show that naively increasing the UTD leads to fast accumulation of bias, but when the bias is properly controlled, we can get huge improvements in sample efficiency.
>
> Throughout the paper, we have deliberately made an effort to keep our algorithm simple and refrained from using complicated hacks to boost performance. As discussed in the paper, a number of decisions are made to ensure our comparisons are meaningful and fair. This simplicity is crucial for making clean analysis and ablations.
>
> Thanks for pointing out that our work can benefit from more theoretical results. The revised submission will include two additional theoretical results: (1) We will provide an outline of the proof of convergence of the tabular form of REDQ when the Robbins-Monroe conditions for the step sizes are satisfied. An outline is sufficient, since the proof is very similar to other such convergence proofs in the literature (for example, in the Maxmin paper). (2) We will derive a bound for the variance of the target for the weighted version of REDQ (i.e., when the target is the expectation of across all samples of size M from the N Q functions). We also show that this bound goes to zero as N goes to infinity.
>
> Thank you for pointing out the latex issues, we will fix them in our revision!

---

### Official Review · AnonReviewer1 · 2020-10-26

**Rating:** 6
**Confidence:** 3

**Review:**

Summary: This paper proposes a new model-free algorithm called Randomized Ensemble Double Q Learning (REDQ) for the optimal control problem. Also, this paper empirically shows that the performance of REDQ is not worse than the model-based algorithm, MBPO.

Comments:
Pro.
1. Empirical studies significantly show that REDQ takes less environment interactions but achieves much higher average return compared to SAC algorithm. Also, the performance of REDQ is even better than MBPO in the Hopper problem and Humanoid problem.
2. The theoretical analysis introduces the relation among two hyper-parameters (M, N) and the expected random approximation error.

Con.
1. In Section 3.1 Theoretical Analysis, the theoretical result is not complete enough. The Maxmin Q-Learning paper, Lan et al. (2020), also proves that Maxmin Q-learning algorithm has a vanishing approximation variance (with N tends to infinity). This result is vaguely mentioned below Theorem 1 but there is no rigorously proof provided.
2. The authors claim that the REDQ algorithm is at least no worse than MBPO. This statement is based on the empirical studies or intuitive explanations rather than theoretical analysis. It will be better to provide more comprehensive analysis of the algorithm.

%--------------------------------%
I thank the authors for clarifying my questions and concerns. The authors have included further theoretical developments in the revision, and they look satisfactory to me. Overall, I tend to accept this paper.

---

> ### Author Response · Authors · 2020-11-18
> **Reply to AnonReviewer1**
>
> Thank you for your mostly positive feedback!
>
> Thanks for pointing out that our work can benefit from more theoretical results. The revised submission will include two additional theoretical results: (1) We will provide an outline of the proof of convergence of the tabular form of REDQ when the Robbins-Monro conditions for the step sizes are satisfied. An outline is sufficient, since the proof is very similar to other such convergence proofs in the literature (for example, in the Maxmin paper). (2) We will derive a bound for the variance of the target for the weighted version of REDQ (i.e., when the target is the expectation across all samples of size M from the N Q functions). We also show that this bound goes to zero as N goes to infinity. This result parallels the variance result in the Maxmin paper, although it is different since the algorithm is different. We also use different tools since we are making much weaker assumptions about the noise term.
>
> For DRL algorithms with non-linear neural networks, it can be challenging to prove mathematically that one method is better than another. But it is certainly an interesting direction for future work.

---

### Official Review · AnonReviewer3 · 2020-10-26
**Good idea and well written**

**Rating:** 7
**Confidence:** 3

**Review:**

Summary
The paper proposes three techniques that altogether greatly improves the performance of soft actor-critic (SAC), resulting in a new algorithm called REDQ. (1) A higher update-to-data ratio, which speeds up the critic update. (2) Using the average ensemble Q for the policy gradient, therefore reducing its variance. (3) Taking the min of a small subset of the ensemble Qs to compute the target Q, therefore reducing the Q bias. The paper also performs extensive ablation studies to prove the importance of each technique.

Recommendation
Overall I think the paper deserves an acceptance. The proposed solution is simple, effective, and justified.

Strengths
1. REDQ is simple and effective. Implementation requires little change to the backbone of SAC. The authors also provide example code. Fig 1 shows clear advantage over SAC.
2. Ablation studies are rigorous: e.g. Fig 3 even studies fractional M = 2.5, when M = 3 is clearly under-performing.
3. The paper is very well written. In particular, the most important algorithm block and experimental results are shown early, leading to a smooth reading experience.

Weaknesses
1. The paper analyzes the normalized standard deviation of Q bias, but in fact the gradient of Q against actions is more important (line 12, Alg 1). It would be great to see analysis on the gradient as well.

2. Some (minor) missing studies.
(a). I am surprised that “Weighted” performs much worse than REDQ  on Ant (Fig 3g). The authors suggest overfitting as a potential cause. But I am curious whether exploration is the real issue, and whether the same happens on other envs.
(b). Algorithm 1 suggests training Q G times before training the policy once. Does training the policy more frequently help? It makes sense since the avg Q has lower variance.
(c). Why not train up to 1, 3, 10 million steps as in the SAC paper (Fig 1)? Especially for Humanoid, The performance isn’t near 6,000 as reached in the SAC paper.

Other feedbacks
Mujoco envs are rather deterministic and the noise comes from the policy itself. Have you considered other more noisy envs? The ensemble could have a larger impact there.

---

> ### Author Response · Authors · 2020-11-18
> **Reply to AnonReviewer3**
>
> Thank you for your detailed review and positive feedback.
>
> For the “Weighted” variant, we agree that exploration can be an issue. Indeed, the “Weighted” variant tends to give more conservative Q estimates, which might discourage the policy from taking exploratory actions. We will make this more clear in the revision.
>
> The study on more policy updates is in fact already given in the appendix, section F. In summary, the results show that for SAC with high UTD, more policy updates often leads to faster bias accumulation and worse performance. For REDQ, more policy updates leads to slightly better performance in some environments, and slightly worse in others. In general, taking fewer policy updates can stabilize Q values and sometimes boost performance, but when the bias is adequately controlled, training is less affected by additional policy updates.
>
> In terms of the number of training epochs, as discussed in section 2.1, page 3, we are mainly comparing with the MBPO results, many of our hyperparameters are set to be consistent with MBPO for a fair comparison, and all our experiments are run using the same number of epochs, with the same evaluation protocol, as done in the MBPO paper.
>
> Our work is currently focused on the MuJoCo benchmark; however, trying this idea in stochastic environments is certainly an exciting direction for future work.
>
> Thank you for the suggestion on performing an analysis of the gradient of Q against actions. However, we are not quite clear what you are suggesting here. Would you be able to provide more explicit details about what you have in mind?

---

### Official Review · AnonReviewer2 · 2020-10-29

**Rating:** 7
**Confidence:** 3

**Review:**

[Summary]

This paper proposes Randomized Ensembled Double Q-Learning (REDQ), a new model-free RL algorithm that aims to improve the sample efficiency over existing model-free methods. Experiments on Mujoco show that REDQ achieves better sample efficiency than popular model-free methods such as SAC and is comparable with model-based methods such as MBPO. The paper further provides extensive ablation studies that justify the necessity of the algorithmic components in REDQ and show that improved Q estimation bias may have been the key reason for the performance gain. The paper also provides some theoretical analysis of the Q estimation bias.

[Pros]

--- The empirical performance of REDQ seems rather strong: significantly better than SAC and can match or exceed MBPO depending on the environment.

--- Section 3 provides a rather convincing explanation of the performance gain through the perspective of the Q estimation bias: REDQ manages to achieve a low bias in terms of both average and std across (s,a) pairs. In comparison SAC fails on both grounds, while AVG (naive ensemble without the in-target minimization) achieves a low std but still a rather high average. The theoretical analysis (Theorem 1) also helped improve my understanding on this front by illustrating how the factors (M, N) could affect the bias.

--- The ablation study in Section 4 is very detailed and answers a lot of questions I had (e.g. importance of M, N) and I liked it a lot.

--- Overall the paper is quite clearly written and conveys the message quite clearly.

[Cons, and comments]

--- My main concern is that the most similar approach Maxmin (Lan et al. 2020) which the authors cited multiple times was not comprehensively tested in the experiments. More concretely, Maxmin was not presented in the main plots (Figure 1 and 2) and only showed up (and in a modified fashion) as an ablation point in Figure 3 where it seemed like its performance was pretty bad. From the original Maxmin paper it seemed like they did not try Mujoco; is it the case that Maxmin did not really scale up to Mujoco?

Specifically, it is a bit disturbing that Maxmin did not appear in Figure 2 which studied the Q estimation bias. Compared with the baselines in that figure (SAC20, AVG), Maxmin sounds much better in terms of reducing the Q estimation error (it also has both ensemble and in-target minimization). I am quite curious how Maxmin does on the bias, and specifically if it does well on the bias but performs worse than REDQ, what is really going on.


--- The key algorithmic novelty of REDQ seems to be the combination of two existing ideas: an ensemble of N=10 networks, as well as in-target minimization over only M=2 randomly sampled networks from the ensemble. I am not entirely sure whether this could be considered as of enough novelty in this area (I am not super familiar with the literature here, so authors / other reviewers please feel free to correct me if I am wrong.) Also, given that the ablation studies showed that having both parts (large N, small M) is indeed important, at this moment I am only thinking of this as a weak concern.

------
After rebuttal: Thank the authors for their efforts in the rebuttal and revision. The authors' response to the Maxmin question (along with the revised discussions) sound convincing to me. I would like to keep my original evaluation and would lean towards acceptance for this paper.

---

> ### Author Response · Authors · 2020-11-18
> **Reply to AnonReviewer2**
>
> Thank you for your very detailed review and your positive feedback.
>
> We would like to first respond to your main concern about Maxmin. We have actually tested multiple Maxmin variants with different ensemble sizes for the MuJoCo environment. The results are very similar to when we change the in-target minimization parameter M in the ablation section: when we increase the ensemble size to be larger than 3, Maxmin starts to reduce the bias so much that we get a highly negative Q bias, and this negative bias accumulates so fast that the Q network becomes unstable, resulting in poor performance. A short discussion on this has been given on page 7. In our revision, we will modify this section to make it more clear.
>
> In the Maxmin paper, the Maximin algorithm was mainly tested on Atari environments with a small finite action space, in which case it provides good performance. Our results show that when using environments with high-dimensional continuous action spaces, such as MuJoCo, the rapid accumulation of (negative) bias becomes a problem. This result parallels some recent research in offline (i.e., batch) DRL. In the paper An Optimistic Perspective on Offline Reinforcement Learning (Agarwal et al., 2019), it is shown that with small finite action spaces, naive offline training with DQN only slightly reduces performance. However, continuous action Q-learning based methods (such as DDPG, SAC) suffer much more from Q bias accumulation compared to discrete action methods. For example, see Off-Policy Deep Reinforcement Learning without Exploration (Fujimoto et al., 2018) and Stabilizing Off-Policy Q-Learning via Bootstrapping Error Reduction (Kumar et al., 2019) for the continuous action case, where naive offline training can entirely fail.
>
> For your concern on the novelty of the paper, a major novelty of our work is a focused study on the high update-to-data (UTD) ratio setting. We show that naively increasing this ratio will lead to high bias and poor performance, but by properly controlling the Q bias, model-free methods can benefit hugely from high UTD. (Additional experiments on different UTDs are presented in appendix section G.) This issue is very important yet rarely studied in the model-free DRL literature. Although our REDQ algorithm seems simple, simplicity is a major strong point. As discussed in the paper, we have deliberately made an effort to keep our algorithm simple, and refrained from using complicated hacks to boost performance, allowing for a fair comparison with other algorithms. Having a simple structure also makes our ablations cleaner, and makes it easier to apply REDQ to other existing algorithms.

---

### Author Response · Authors · 2020-11-18
**Response to All Reviewers**

We would like to thank all the reviewers for their detailed reviews and positive feedback. In response to your insightful comments, we are preparing a minor revision of our original submission. We will upload the revision before the end of the rebuttal period.

In our paper, we introduce a novel model-free algorithm, Randomized Ensembled Double Q-Learning (REDQ), and show that its performance is just as good as, if not better than, the state-of-the-art model-based algorithm for the MuJoCo benchmark. This result indicates that, at least for the MuJoCo benchmark, models may not be necessary for achieving high sample efficiency. Moreover, REDQ can achieve this performance using fewer parameters than the model-based method, and with less wall-clock run time. REDQ also provides a huge improvement in sample efficiency over SAC.

We defined the Update-To-Data (UTD) ratio as the number of updates taken by the agent compared to the number of actual interactions with the environment. Most of the well-known model-free algorithms (SAC, DDPG and so on) use a UTD ratio equal to one; when higher UTD ratios are used, performance declines dramatically. REDQ, on the other hand, is able to use a UTD ratio >> 1 and achieve high sample efficiency and high performance. This is achieved by carefully integrating double Q-learning with an ensemble framework. To our knowledge, for continuous action spaces, REDQ is the first successful model-free DRL algorithm using a UTD ratio >> 1. Our paper provides a comprehensive analysis through mathematical arguments as well as extensive experimental and ablation studies.

We believe that the paper is important for the community. In order to improve the paper’s visibility, if you are satisfied with our response and revision, we kindly ask you to consider increasing your score and recommending acceptance as a spotlight paper. Thank you.

---

### Author Response · Authors · 2020-11-25
**Revision has been uploaded**

We would like to thank the reviewers again for your insightful comments. We have just uploaded our revision. Here is a summary of changes:

In response to Reviewer2, We now have a more extensive discussion on why Maxmin does not work well on the continuous action space MuJoCo environments with large ensemble sizes. In section 3 (top of page 5) we mentioned the explanation is given in section 4. In section 4 (variants and ablations, page 7-8), we now have a more clear and detailed discussion. We also cited recent works in offline (batch) DRL that might provide more insight for the readers.

Following the suggestions of Reviewer 1 and Reviewer 4, we added two additional theoretical results to section 3 of the main paper (page 6-7), and section A of the appendix. We derive a bound for the variance of the target for the weighted version of REDQ, and provide a proof of convergence for REDQ in the tabular case.

Following the suggestions of Reviewer3, we added comparison of REDQ and the Weighted variant on all four environments to appendix H. Results show that overall REDQ is more robust, though their performance is similar in environments other than Ant. We cited additional works on exploration to help explain the difference in performance.

We have also fixed some minor issues and typos.

---

### Decision · Program_Chairs · 2021-01-07
**Final Decision**

**Decision:**

Accept (Poster)

**Comment:**

This paper develops an effective model-free algorithm that achieves high sample efficiency. The empirical performance is appealing, which is comparable to model-based policy optimization and significantly outperforms SAC. The paper is well-written, and contains rigorous ablation studies. Weakness: the theoretical analysis is Section 3.1 is not thorough yet,  and it would be helpful to include more numerical comparisons with the Maxmin approach by Lan et al.